# Self-supervised Training of Proposal-based Segmentation via Background Prediction

## Abstract

While supervised object detection and segmentation methods achieve impressive accuracy, they generalize poorly to images whose appearance significantly differs from the data they have been trained on. To address this in scenarios where annotating data is prohibitively expensive, we introduce a self-supervised approach to detection and segmentation, able to work with monocular images captured with a moving camera. At the heart of our approach lies the observations that object segmentation and background reconstruction are linked tasks, and that, for structured scenes, background regions can be re-synthesized from their surroundings, whereas regions depicting the object cannot. We encode this intuition as a self-supervised loss function that we exploit to train a proposal-based segmentation network. To account for the discrete nature of the proposals, we develop a Monte Carlo-based training strategy that allows the algorithm to explore the large space of object proposals. We apply our method to human detection and segmentation in images that visually depart from those of standard benchmarks, achieving competitive results compared to the few existing self-supervised methods and approaching the accuracy of supervised ones that exploit large annotated datasets.

## 1 Introduction

Recent object detection and segmentation methods have reached impressive precision and recall rates when trained and tested on large annotated datasets (Lin et al., 2014). However, large and varied datasets do not warrant the best possible performance in a particular application domain, as each comes with its own challenges and opportunities. While domain-specific models are thus needed, annotating separate and sufficiently large datasets in all scenarios is impractical.

Therefore, weakly- and self-supervised detection and segmentation of salient foreground objects in complex scenes have recently gained attention Croitoru et al. (2018); Eslami et al. (2016); Crawford & Pineau (2019); Rhodin et al. (2019); Bielski & Favaro (2019). These methods promise effortless processing of community videos with little human intervention. However, a closer look at existing techniques reveals that they make strong assumptions, such as the target objects being on top of a static background, or rely on pre-trained supervised object localization, object-boundary detection, and optical flow networks. This severely limits their applicability in practice.

To develop a more generic technique, we start from the observation that in most images the background forms a consistent, natural scene. Therefore the appearance of any background patch can be predicted from its surroundings. By contrast, a salient object's appearance is unpredictable from the neighboring scene content and can be expected to be very different from what an inpainting algorithm would produce. We incorporate this insight into a proposal-generating deep network whose architecture is inspired by those of YOLO Redmon et al. (2016) and MaskRCNN He et al. (2017) but does not require explicit supervision.

For each proposal, we synthesize a background image by masking out the corresponding region and inpainting it from the remaining image. The loss function we minimize favors the largest possible distance between this reconstructed background and the input image. This encourages the network to select regions that cannot be explained from their surrounding and are therefore salient. To handle the discrete nature of the proposals, we introduce a Monte Carlo-based strategy to train our network. It operates on a discrete distribution, is unbiased, exhibits low variance, and is end-to-end trainable.

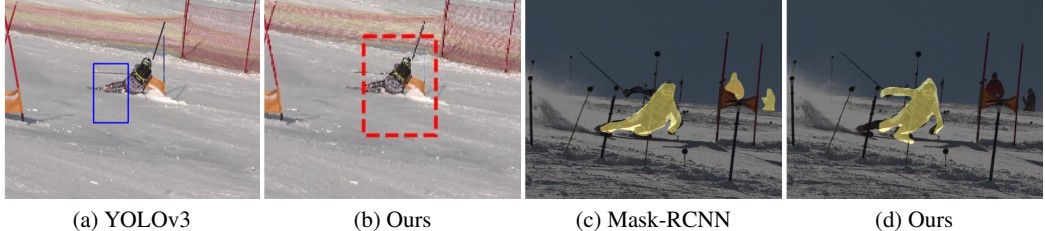

|  (a) YOLOv3 | (b) Ours | (c) Mask-RCNN | (d) Ours |

Figure 1: **Domain specific detection and segmentation.** Our self-supervised method detects the skier well, while YOLO trained on a general dataset does not generalize to this challenging domain. Similarly, MaskRCNN trained on a general dataset sometimes misses body parts and detects people in the background.

We tailor our training algorithm to segmenting human motion from videos in which a single subject moves. The constraints imposed by this well-defined, yet challenging and widely studied task, allow us to generalize to arbitrary, dynamic camera motion. As a consequence, it lets us perform end-to-end training without the bootstrapping Croitoru et al. (2019) and intermediate optical flow supervision Papazoglou & Ferrari (2013); Jain et al. (2017); Tokmakov et al. (2017).

We demonstrate the effectiveness of our unsupervised method on several datasets captured with increasingly mobile cameras, ranging from static to pan-tilt-zoom and hand-held. We will show that our approach applies to images acquired in conditions significantly more general than those of standard benchmarks, without requiring *any* manual annotation. Thus, as shown in Fig. 1, it approaches the quality and sometimes outperforms state-of-the-art detectors that have been trained on large annotated datasets. Retraining or fine-tuning these methods on this data could be done but would require supervision that is hard to obtain, which makes a self-supervised approach attractive. We will make our code and ground-truth segmentations for the existing **Ski-PTZ-Dataset** and the new **Handheld190k** dataset publicly available upon acceptance of the paper.

## 2 RELATED WORK

Most salient object detection and segmentation algorithms are fully-supervised (Redmon et al., 2016; He et al., 2017; Song et al., 2018; Cheng et al., 2017) and require large annotated datasets with paired images and labels. Our goal is a purely self-supervised method that succeeds without segmentation and object bounding box annotations. Note that this differs from the so-called *unsupervised object detection* methods (Hu et al., 2018; Li et al., 2018; Jain et al., 2017), which still require domain-specific annotations at training but not at test time. We focus our discussion on self- and weakly-supervised methods with regard to the type of training data used and refer to (Koh & Kim, 2017) for the discussion of methods using hand-crafted optimization.

**Weakly-supervised methods.** A classical weakly-supervised example is the Hough Matching algorithm (Cho et al., 2015). It uses an object classification dataset and identifies foreground as the image regions that have re-occurring Hough features within images of the same class. Similar principles have been followed using deep networks trained for object detection (Wei et al., 2017; Jain et al., 2017), optical flow (Tokmakov et al., 2017), and object saliency (Li et al., 2018). These methods make the implicit assumption that the background varies across examples and can therefore be excluded as noise. This assumption is violated in the targeted case of training on domain-specific images, where foreground and background are similar across examples.

**Motion-based methods** Given video sequences, the temporal information can be exploited by assuming that the background changes slowly (Barnich & Van Droogenbroeck, 2011) and linearly (Stretcu & Leordeanu, 2015). However, even a static scene induces non-homogeneous deformations under camera translation, and it can be difficult to handle all types of camera motion (pan, tilt, zoom) within a single video, and distinguish articulated human motion from background motion (Russell et al., 2014). Some of the resulting errors can be corrected by iteratively refining the crude background subtraction results of Stretcu & Leordeanu (2015) with an ensemble of student and teacher networks (Croitoru et al., 2018; 2019). This, however, induces a strong dependence on the teacher used for bootstrapping.

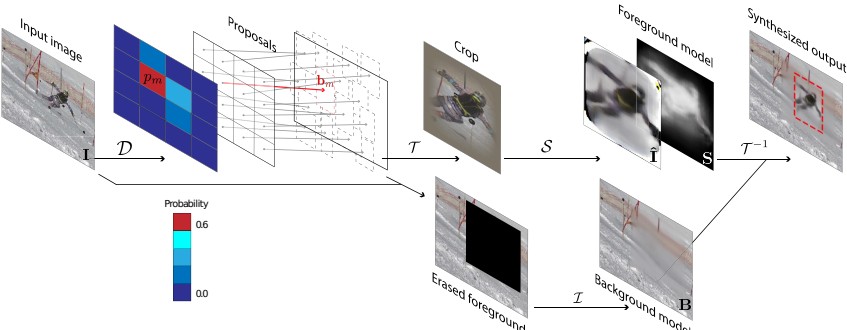

Figure 2: **Method overview.** Encoder-decoder network ($\mathcal{S}$) with an attention mechanism defined by proposal-based detection ($\mathcal{D}$) and spatial transformers ($\mathcal{T}, \mathcal{T}^{-1}$). Via the use of an inpainting network ($\mathcal{I}$), our approach makes it possible to train this network in an entirely self-supervised manner on unknown scenes with a moving background and captured with a hand-held camera.

Our approach is conceptually closely related to VideoPCA (Stretcu & Leordeanu, 2015), which models the background as the part of the scene that can be explained by a low-dimensional linear basis. This implicitly assumes that the foreground is harder to model than the background and can therefore be separated as the non-linear residual. Here, instead of using motion cues, we propose to rely on the predictability of image patches from their spatial neighborhood.

**Self-supervised Methods.** Most similar to our approach are the self-supervised ones to object-detection (Eslami et al., 2016; Crawford & Pineau, 2019; Rhodin et al., 2019; Bielski & Favaro, 2019) that complement auto-encoder networks by an attention mechanism. These methods first detect one or several bounding boxes whose content is extracted using a spatial transformer (Jaderberg et al., 2015). This content is then passed through an auto-encoder and re-composited with a background. In (Rhodin et al., 2019), the background is assumed to be static and in (Eslami et al., 2016; Crawford & Pineau, 2019) even single colored, a severe restriction in practice. Crawford & Pineau (2019) use a proposal-based network similar to ours, but resort to approximating the proposal distribution with a continuous one to make the model differentiable. Here, we demonstrate that much simpler importance sampling is sufficient. In (Bielski & Favaro, 2019), a generative model relies on the assumption that the image region strictly covering the salient object can be subject to random shifts without affecting the realism of the scene. However, this method can easily be deceived by other background objects whose random displacement can still yield realistic images. By contrast to all of these methods, our approach works with images acquired using a moving camera and given an arbitrarily colored background.

In addition to object detection, the algorithm of Rhodin et al. (2019) also returns instance segmentation masks by reasoning about the extent and depth ordering of multiple people in a multi-camera scene. However, this requires multiple static cameras and a static background at training time, as does the approach of Baqué et al. (2017) that performs instance segmentation in crowded scenes.

## 3 METHOD

Our goal is to learn a salient person detector and segmentor from unlabeled videos acquired in as generic a setup as possible, including using hand-held cameras. At inference time, our algorithm takes a single image $\mathbf{I} \in \mathbb{R}^{W \times H}$ as input and outputs a bounding box $\mathbf{b}_m \in \mathbb{R}^4$, expressed as a center location, width, and height, and a segmentation mask $\mathbf{S} \in \mathbb{R}^{128 \times 128}$ within that window. This is achieved by the multi-stage process visualized in Fig. 2. A first network predicts a set of $C$ candidate object locations $(\mathbf{b}_c)_{c=1}^C$ and corresponding probabilities $(p_c)_{c=1}^C$. To this end, we use a fully-convolutional architecture that divides the image into a grid, with each cell $c$ yielding one probability, $p_c$, and an offset from the cell center to compute $b_c$, similarly to YOLO Redmon et al. (2016). Second, the $\mathbf{b}_m$ with highest probability $p_m \geq p_c$ , $\forall c$, is chosen and its content is decoded into foreground $\hat{\mathbf{I}} \in \mathbb{R}^{128 \times 128}$, segmentation mask $\mathbf{S}$, and background $\mathbf{B} \in \mathbb{R}^{W \times H}$ with separate encoder-decoder branches.

### 3.1 SELF-SUPERVISED TRAINING WITH KNOWN BACKGROUND

Given a set of unlabeled images $(\mathbf{I}_i)_{i=1}^N$, our goal is to train a neural network $\mathcal{D}(\mathbf{I}) \mapsto ((\mathbf{b}_c), (p_c))_{c=1}^C$ to propose suitable bounding box candidates and the corresponding probabilities to select the $\mathbf{b}_m$ that contains an object. Because object locations are unknown, we do this with an autoencoder objective, $L(\mathcal{F}(\mathbf{I}, \mathbf{b}_c, \mathbf{B}), \mathbf{I})$, that measures how well the autoencoder $\mathcal{F}$ reproduces the input image $\mathbf{I}$ on top of a background $\mathbf{B}$, with the attention on $\mathbf{b}_c$. As in (Crawford & Pineau, 2019; Rhodin et al., 2019), this is implemented with a spatial transformer $\mathcal{T}$ that crops the area of interest, followed by a bottle-neck autoencoder $\mathcal{S}$ that produces the foreground and segmentation mask, and a second spatial transformer $\mathcal{T}^{-1}$ that undoes the cropping and blends the synthesized foreground and background. We write

$$\mathcal{F}(\mathbf{I}, \mathbf{b}_c, \mathbf{B}) = \mathcal{T}^{-1}(\hat{\mathbf{I}}) \circ \mathcal{T}^{-1}(\mathbf{S}) + \mathbf{B} \circ (1 - (\mathcal{T}^{-1}(\mathbf{S})), \text{ with } \mathcal{S}(\mathcal{T}(\mathbf{I}, \mathbf{b})) \mapsto (\hat{\mathbf{I}}, \mathbf{S}), \quad (1)$$

where $\circ$ is the elementwise multiplication. The loss $L$ can be a least-square distance between all pixel values, alternatives are discussed at the end of the section. Because the attention window $\mathbf{b}$ selects part of the image for decoding, this loss encourages $\mathcal{D}$ to focus on the object so as to model the foreground on top of $\mathbf{B}$. Moreover, the autoencoder only approximates the actual image, forcing the segmentation mask to contain only the parts not captured by the background.

For now, we assume $\mathbf{B}$ to be known, but we will remove this assumption later. In this setting, we derive a probabilistic formulation in which not a single, but multiple candidates can give rise to plausible reconstructions. We then reason about the expected loss across all likely candidates,

$$O(\mathbf{I}) = \mathbf{E}_c \left[ L(\mathcal{F}(\mathbf{I}, \mathbf{b}_c, \mathbf{B}), \mathbf{I}) \right], \text{ with } c \sim p, \text{where } p(c) = p_c, \quad (2)$$

where $\mathbf{E}_c$ denotes the expectation over $c$ drawn from the categorical proposal distribution output by the network $\mathcal{D}(\mathbf{I})$. We optimize Eq. (2) with stochastic gradient descent and mini-batches of $(\mathbf{I}_i)_{i=1}^N$. Note that minimizing this objective will jointly optimize the detection network $\mathcal{D}$ that generates and stochastically selects proposals, and the synthesis network $\mathcal{S}$ that models the object appearance and the segmentation mask.

Because we have a finite set of candidates, Eq. (2) can be expressed deterministically as a weighted sum over all candidates as

$$O(\mathbf{I}) = \sum_{c=1}^C p(c) L(\mathcal{F}(\mathbf{I}, \mathbf{b}_c, \mathbf{B}), \mathbf{I}). \quad (3)$$

This objective is an explicit function of $p$ and could be optimized by gradient descent. However, this sum over $C$ forward passes is inefficient to evaluate in practice when $C$ is large. For example, in our experiments $C = 64$, which does not fit in memory. This was also observed by Crawford & Pineau (2019), who resorted to using a continuous approximation of the discrete distribution $p$ to facilitate end-to-end training. Here, we propose a simpler alternative, exploiting Monte Carlo and importance sampling, which provides an unbiased estimator with low variance.

**Monte Carlo (MC) sampling.** In principle, the expectation in Eq. (2) could be estimated by sampling a small set of $J$ candidate cells from $p$. Unfortunately, sampling from such a discrete distribution is not differentiable with respect to its parameters, which precludes end-to-end gradient-based optimization. Instead of sampling according to $p$, we can rewrite Eq. (2) using an arbitrary distribution $q$, by reweighting with the quotient of both distributions. That is,

$$O(\mathbf{I}) = \mathbf{E}_c \left[ \frac{p(c)}{q(c)} L(\mathcal{F}(\mathbf{I}, \mathbf{b}_c, \mathbf{B}), \mathbf{I}) \right], \text{ with } c \sim q. \quad (4)$$

This change of distribution and relative weighting holds for any two probability distributions, as explained in the appendix. In practice, we approximate the expectation on a mini-batch, with a single sample drawn from $q$ per image.

**Importance sampling.** While moving the distribution into the expectation sum provides differentiability, it comes with the drawback of a potentially large variance, i.e., high approximation error for few samples. For instance, by choosing $q$ to be the uniform sampling distribution $\mathcal{U}$, most of the uniformly drawn samples will have a low probability in $p$ and therefore negligible influence. To reduce this variance, we leverage importance sampling and set the sampling distribution $q$ to be

similar or equivalent to $p$. To prevent division by very small values that could lead to numerical instability, we define the new distribution $q$ as

$$q(c) = p(c)(1 - C\epsilon) + \epsilon \, . \tag{5}$$

As a side effect, $\epsilon$ controls the probability that an unlikely case is chosen, which induces a form of exploration that is helpful in the early training stages of the network. Note that the fraction $\frac{p(c)}{q(c)}$ cancels numerically for $q(c) = p(c)$. However, while values cancel, their derivatives do not; the differentiability of $p$ is maintained as the sampling distribution $q$ must be treated as a constant. In this case, the gradient of Eq. 4 equals that of the likelihood ratio method (Glynn, 1990) used in the REINFORCE algorithm (Williams, 1992). We favor the MC interpretation that explains the effect of the gradient quotient as importance sampling and generalizes to arbitrary $q$. While the relation is known for a long time (Glynn, 1990), is overlooked in the recent literature.

### 3.2 Self-Supervised Training with Moving Background

Having derived an efficient training scheme for proposal-based segmentation when $\mathbf{B}$ is given, we would like to address the moving camera scenario, where it is not given, by reducing this more difficult case to the former. This can be achieved by predicting the background image, which, in the absence of prior shape and appearance information of the foreground, is even harder than segmentation.

To overcome this difficulty, we redefine the training strategy and replace background prediction with the simpler task of inpainting local regions, which can easily be trained by removing a region and predicting it from its immediate surrounding (Pathak et al., 2016; Yu et al., 2018). Under the assumption that the foreground object moves, a network, $\mathcal{I}$, trained on this self-supervised inpainting task would not manage to reconstruct the foreground objects if fully removed in the input because the surrounding background gives no cues of their presence. We therefore cast foreground segmentation as the task of finding the area $\mathbf{b}_c$ that, when inpainted, yields the largest image reconstruction error.

Instead of doing an expensive search at inference time, we train $\mathcal{D}$ to predict the foreground location. We search probabilistically by sampling likely cases according to the current distribution $p$ and write

$$G(\mathbf{I}) = -\mathbf{E}_c \left[ \frac{L(\mathcal{I}(\mathbf{I}, \mathbf{b}_c), \mathbf{I})}{a(\mathbf{b}_c)} \right], \text{ with } c \sim p, \tag{6}$$

where $\mathcal{I}$ takes the image $\mathbf{I}$ and the region $\mathbf{b}_c$ to inpaint as input, $L$ is a pixel loss, $a(\mathbf{b}_c)$ normalizes the loss by the window area, and $p$ and $\mathbf{b}_c$ are computed with $\mathcal{D}$ as before. We use the same importance sampling strategy to optimize the discrete distribution, but opposed to the foreground objective, we use the negative expectation. This negative reconstruction error encourages the selection of those regions were the true image is dissimilar to the reconstructed background when minimizing Eq. (6).

**Training strategy.** Unfortunately, the objective $G$ has trivial solutions. It favors locations with high error density, irrespectively of their size, as illustrated in Fig. 3(b). Alternatively, removing the normalization by the area favors erasing extensively large regions, containing an object or not, because larger number of reconstructed pixels lead to higher inpainting errors, as shown in Fig. 3(d).

To overcome these degenerate cases, we combine the new background objective $G$ with the foreground objective $O$ of Eq. (4), substituting the known $\mathbf{B}$ with the learned inpainting $\mathcal{I}(\mathbf{I}, \mathbf{b}_c)$. The reason behind this is that these two terms are complementary: While $G$ prefers locations that cover the object neither precisely nor entirely, $O$ favors a tight fit over partial coverage but has a trivial solution when $\mathbf{b}_c$ is on a background region, i.e., not covering the object and having nothing to encode.

Finding a balance between these two adversarial objectives by relative weighting turned out to be difficult, if not impossible. Instead, we separate them such that their influence on the individual network components is mutually exclusive. Probabilities $p_c$ are only optimized according to $G$. Therefore, $O$ can not impose bias to background regions where it has a trivial solution. In turn, $\mathbf{b}_c$ is optimized only by $O$ to find a tight fit without the bias of $G$ to excessively large or small proposals $\mathbf{b}_c$. Also $\mathcal{S}$ is solely optimized by $O$ to output the best possible reconstruction, instead of the largest distance to the background as induced by $G$. The separation can conveniently be computed in a single forward-backwards pass by treating the excluded variables as constants in the respective objectives, that means cutting of gradient flow through them.

Note that the stable training through effective separation into coarse and fine localization of adversarial termsis only possible with the chosen proposal-based detection framework; not with direct regression.

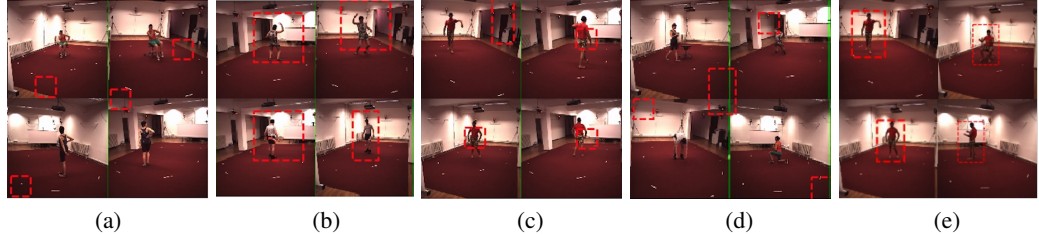

|     |     |     |     |     |
| --- | --- | --- | --- | --- |
| (a) | (b) | (c) | (d) | (e) |

Figure 3: **Ablation study on H36M.** (a) Uniform sampling does not converge. (b) Joint training of $O$ and $G$ (c) only $G$ (d) direct regression of a single bounding box using $O$ and $G$ (e) Ours.

**Implementation details.**    Since no labels for intermediate supervision are available, naive end-to-end training is difficult. To counteract this, we use ImageNet-trained weights for initialization; rely on a perceptual loss on top of the per pixel $L_2$ loss for $L$; exploit the Focal Spatial Transformers (FST) of Rhodin et al. (2019) to speed up convergence; and scale the erased region in $\mathcal{I}$ to be 1.1 the size of that predicted by $\mathcal{D}$ to increase the chances of covering the object. Moreover, we limit the location offsets to 1.5 the cell width and discard those outside the image. In addition, we rely on $L_2$ priors on the output of $\mathcal{D}$, and a v-shaped prior $L_v$ on $\mathbf{S}$ that stabilizes early training iterations by pushing the solution at the beginning of the training to be larger than a value $\lambda$, yet sparse when exceeding this threshold. We performed a grid search on the relative weights of the terms, the offset limits, and $\lambda$. The pixel reconstruction and perceptual losses are weighted 1:2, and the priors have a weight of 0.1, 1, and 0.1, respectively, to compensate for their different magnitudes. Additional details are given in the appendix.

Regarding the inpainting network, using an off-the-shelf inpainting method leads to hallucinated objects. Therefore, we train the inpainting network in a self-supervised manner on the available input videos. Details are given in the appendix, Section A.1.

## 4    EXPERIMENTS

In this section, we evaluate our approach to self-supervised salient object detection and segmentation. Note that our algorithm works on single images at inference time and requires the background inpainting model only at training time. Our focus is on people-detection. Nevertheless the method could generalize to arbitrary objects, as long as videos or picture collections of a single object in front of the same scene are available. We first demonstrate the use of our method in a controlled environment with static background, to compare it to a state-of-the-art self-supervised approach. Then, we introduce skiing footage acquired using PTZ-cameras and footage of people performing 14 everyday activities recorded using hand-held cameras to demonstrate that existing supervised methods that do well in the controlled environment struggle to adapt to such challenging conditions, whereas our approach delivers promising results. We provide additional qualitative results, experiments on the loss function, and details on the parameter search in the appendix material.

### 4.1    PEOPLE IN A CONTROLLED ENVIRONMENT

We compare our method against state-of-the-art ones on the **Human3.6m** dataset (Ionescu et al., 2014) that comprises 3.6 million frames and 15 motion classes. It features 5 subjects for training and 2 for validation, seen from different viewpoints against a static background and with good illumination.

**Comparative Results.**    On the left side of Table 1, we compare our detection accuracy to that of a very recent self-supervised deep learning method (Rhodin et al., 2019), using the mean detection precision (mAP), the mean precision of having an intersection-over-union (IoU) of more than 50%. Our slightly lower accuracy stems from not explicitly assuming a static background, which Rhodin et al. (2019) do. While valid in a lab, this assumption results in total failure in outdoor scenes with moving backgrounds, such as those discussed the next sections.

**Ablation Study.**    Here we show that our model choices for training and probabilistic inference are important. Using uniform sampling instead of importance sampling, does not converge, as shown in Fig. 3(a). Fig. 3(b) illustrates that joint training of $\mathcal{D}$ with $O$ and $G$, instead of our separate one, produces bounding boxes that are too large. Fig. 3(c) shows that using only the background objective leads to small detections that miss the subject and (d) that direct regression without multiple

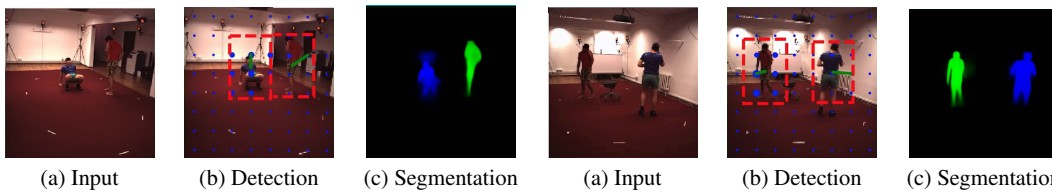

| (a) Input | (b) Detection | (c) Segmentation | (a) Input | (b) Detection | (c) Segmentation |

Figure 4: **Multi-person detection and segmentation results**, generated by sampling our model multiple times. As the model is trained on single persons this only works for non-intersecting cases.

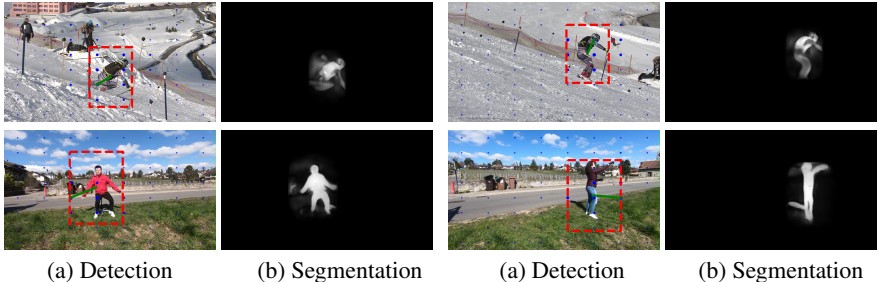

| (a) Detection | (b) Segmentation | (a) Detection | (b) Segmentation |

Figure 5: **Qualitative results on Ski-PTZ-camera and Handheld190k.** Example results on the test images. (a) The detection results show the predicted bounding box with red dashed lines, the relative confidence of the grid cells with blue dots and the bounding box center offset with green lines. (b) Soft segmentation mask predictions. Note that in the second row, the moving clouds are not segmented but the shadow of the person can be included.

candidates diverges. These failure cases are representative of the behavior on the whole dataset. Fig. 3(e) demonstrates that our full model using the separate training strategy and importance sampling can accurately detect the person and estimate tighter bounding boxes.

**Multiple people.** Although our focus is on handling single objects or persons, our probabilistic framework can handle several at test time by sampling more than once. Fig. 4 shows the predicted cell probability as blue dots whose size is proportional to the probability. The fully-convolutional architecture operates locally and thereby predicts a high person probability close to both subjects. As a result, both the detection and segmentation results remain accurate as long as the individuals are sufficiently separated.

### 4.2 SKIERS FILMED USING A PTZ-CAMERAS

We now turn to the out-of-the-ordinary motions of six skiers on a slalom course featured in the **Ski-PTZ-Dataset** (Rhodin et al., 2018b). The six skiers are split four/one/one to form training, validation, and test sets, with, respectively, 7800, 1818 and 1908 frames. The intrinsic and extrinsic parameters of the pan-tilt-zoom cameras are constantly adjusted to follow the skier. As a result, nothing is static in the images, the background changes quickly, and there are additional people standing as part of the background. We use the full image as input, evaluate detection accuracy in relation to the available 2D pose annotation, and segmentation accuracy by manually segmenting 9 frames from each of the six cameras which add up to 108 frames for two test sequences. For the hyperparameter selection we use the manually segmented 3 frames from each of six cameras which is in total 36 frames for two validation sequences.

In Table 1, we evaluate detection accuracy. Note that our method delivers an $mAP_{0.5}$ score that is significantly better than that of the general YOLO Redmon et al. (2016) detector trained on MSCOCO.

In Table 2, we compare our approach to several state-of-the-art segmentation baselines in terms of precision, recall, F-, and J-measure as defined in (Pont-Tuset et al., 2017). To be fair, we compensate for different segmentation masks quantification levels by a grid search (at 0.05 intervals) to select the best threshold in terms of J-measure for each method. Interestingly, MaskRCNN trained on a large generic dataset is outperformed by ARP on this dataset. Without using any object localization data, our method is on par with MaskRCNN and close to weakly supervised methods that train on large datasets with motion boundary and segmentation mask annotations. More importantly, our approach

(a) Bielski & Favaro (2019)                              (b) Ours

Figure 6: **Reconstructed scene and mask generated by Bielski & Favaro (2019) and our method on training examples.** On the **Ski-PTZ-Dataset** the poles and snow patches are segmented as foreground. On the **Handheld190k** dataset, the masks contain the foreground subject together with the ground they are standing on.

| **H36M** dataset | | **Ski-PTZ-Dataset** | |
|---|---|---|---|
| Method | mAP$_{0.5}$ | Method | mAP$_{0.5}$ |
| NSD Rhodin et al. (2019) | **0.710** | YOLOv3 Redmon et al. (2016) | 0.155 |
| Ours | 0.580 | Ours | **0.278** |

Table 1: **Detection** results on the **H36M** and **Ski-PTZ-Dataset** datasets. They are expressed in terms of mAP$_{0.5}$, the mean probability of having an intersection-over-union (IOU) of more than 50%.

outperforms the self-supervised one of Stretcu & Leordeanu (2015). While our F- and J-measures are slightly lower than that of (Croitoru et al., 2019), part of this difference can be attributed to Croitoru et al. (2019) using a segmentation mask discriminator that is trained on the combination of the ImageNet VID and YouTube Objects datasets. Albeit also trained in a self-supervised fashion, it thereby leverages additional information.

In Fig. 6, we show the scenes reconstructed by Bielski & Favaro (2019) along with the resulting segmentation masks. Note that their generative model fails to segment the foreground object alone and instead segments background objects and sometimes even the ground. This method relies on the property that foreground regions can undergo random perturbations without altering the realism of the scene. However, in the **Ski-PTZ-Dataset**, some background objects, such as poles, also satisfy this property, and the generator can choose to keep these regions. Since the results on the training samples are very poor, we do not provide any quantitative results of this work on the test data.

Further qualitative results are shown in Fig. 5. The probability distribution, visualized as blue dots that increase in magnitude with the predicted likelihood, show clear peaks on the persons. The limitations include slightly blurred and bleeding masks and occasional false positives, reducing precision.

We investigate the effectiveness of different mask priors and ImageNet pre-training on the validation part of the **Ski-PTZ-Dataset** in the appendix. Our $\ell_v$ prior achieves the highest scores in all measures with consistently reliable results. They demonstrate that imposing regularization on the segmentation masks allows us to obtain sharper masks, removing the noise around the foreground object.

### 4.3 DAILY ACTIVITIES CAPTURED USING HAND-HELD CAMERAS

We introduce a new **Handheld190k** dataset that features three training, one validation and one test sequences, comprising $120\,855$, $23\,076$ and $46\,326$ images, respectively, with a single actor performing actions mimicking those in **H36M**. We manually annotated 112 frames in the validation and 240 frames in the test sequence to provide ground-truth segmentation masks which we believe will be useful for evaluating other self- and weakly-supervised methods. The camera operators moved

| Method | **Ski-PTZ-Dataset** | | | | **Handheld190k** | | | |
|---|---|---|---|---|---|---|---|---|
| | Precision | Recall | F Measure | J Measure | Precision | Recall | F measure | J measure |
| MaskRCNN He et al. (2017)[d] | 0.75 | 0.65 | 0.68 | 0.65 | **0.93** | **0.88** | **0.90** | **0.77** |
| ARP Koh & Kim (2017)[w] | **0.94** | 0.76 | **0.83** | **0.73** | 0.91 | 0.64 | 0.74 | 0.67 |
| VideoPCA Stretcu & Leordeanu (2015)[s] | 0.49 | **0.84** | 0.61 | 0.56 | 0.35 | **0.87** | 0.49 | 0.47 |
| Unsup-DilateU-Net Croitoru et al. (2019)[s] | 0.74 | 0.76 | **0.74** | **0.65** | **0.81** | 0.72 | 0.75 | 0.67 |
| Ours[s] | **0.75** | 0.56 | 0.63 | 0.56 | 0.80 | 0.74 | **0.76** | **0.68** |

Table 2: **Segmentation** results on the **Ski-PTZ-Dataset** and **Handheld190k** dataset. Ours exceeds or is on par with the self-supervised methods (marked with [s]), and approaches the accuracy of weakly-supervised (marked with [w]) and fully-supervised methods (marked with [d]).

laterally, to test robustness to camera translation and hand-held rotation. We provide examples of our detection and segmentation results in Fig. 5, more are given in the appendix. Our method is robust to the undirected camera motion and to dynamic background motion, such as branches swinging in the wind and clouds moving, and to salient textures in the background, such as that of the house facade.

To perform a quantitative comparison, we use 240 manually segmented test images taken from different motion classes with the subject in many different poses. We evaluate against the same methods using the same quantities as for skiing in Table 2. In this scenario, MaskRCNN yields the highest scores, which is not surprising as the tested sequences are similar to its training set MSCOCO. It is followed by our work in F- and J-measure. Despite using a discriminator that is trained in an unsupervised fashion on another, larger dataset, (Croitoru et al., 2019) performs comparably to ours. Contrary to **Ski-PTZ-Dataset** results, we outperform ARP Koh & Kim (2017) in F- and J-measure and have a larger margin on VideoPCA Stretcu & Leordeanu (2015). Both of these methods often fail in separating the non-homogeneously moving background due to the hand-held camera motion.

## 5  CONCLUSION

We have proposed a self-supervised method for object detection and segmentation that lends itself for application in domains where general purpose detectors fail. Our core contributions are the Monte Carlo-based optimization of proposal-based detection, new foreground and background objectives, and their joint training on unlabeled videos captured by static, rotating and handheld cameras. Our experiments demonstrate that, even if trained only on single persons, our approach generalizes to multi-person detection, as long as the persons are sufficiently separated. In contrast to many existing solutions Barnich & Van Droogenbroeck (2011); Russell et al. (2014); Croitoru et al. (2018), our approach does not exploit temporal cues. In the future, we will integrate temporal dependencies explicitly, which will facilitate addressing the scenario where multiple people interact closely, by incorporating physics-inspired constraints enforcing plausible motion.

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

## A  APPENDIX

In this document, we supply additional implementation details, qualitative results, and a mathematical justification for the importance sampling transformations.

### A.1  IMPLEMENTATION DETAILS

For efficiency, $\mathcal{W}$ and $p$, and $\mathcal{I}$ and $\mathcal{S}$ share parameters, except for their output layers. In the following, we explain the architecture of each sub-network.

**Detection network.**  We predict one candidate relative to each cell in a regular grid using a fully-convolutional architecture similar to YOLO Redmon et al. (2016). We use a ResNet-18 backbone He et al. (2016), that reduces the input dimensionality by a factor 8, forming a low resolution grid of features, e.g., to spatial resolution $8 \times 8$ from $128 \times 128$. The feature size is set to 5; two for bounding box location, two for scale, and one for the probability. Each feature output represents one cell, the offset is predicted relative to the cell center as sketched in Fig. 2 (main document). The output of $p$ is forced to be positive and form a proper distribution, with $\sum_{c=1}^{C} p_c = 1$, by a soft-max activation unit. To prevent this network from constantly predicting bounding boxes at the borders of the image, we zero the outer cell probabilities.

**The synthesis network.**  $\mathcal{S}$ has the form of a bottle-neck auto-encoder, based on the publicly available implementation of Rhodin et al. (2018a). The encoding part is a 50-layer residual network and weights are initialized with ones trained on ImageNet classification. The hidden layer is 728 dimensional, split into an 600 dimensional space that goes through two fully connected layers and a 128 dimensional space that is replicated spatially to an $128 \times 8 \times 8$ feature map to encode spatially invariant features. The decoding is done with the second half of a U-Net architecture Ronneberger et al. (2015) with 32, 64, 128, 256 feature channels in each stage. The final network layer outputs four feature maps, three to predict the color image $\hat{\mathbf{I}}$ and one for the segmentation mask.

**Discussion**  We use a YOLO detection architecture combined with spatial transformers and syntesis network $\mathcal{S}$. To improve inference speeds, one could merge $\mathcal{D}$ and $\mathcal{S}$ to share the same base network through RoI pooling as in MaskRCNN. Nevertheless, the MC training approach is necessary, as the inpainting network $I$ can not share the same features for multiple bounding box locations since the blacked-out input region varies for each $\mathbf{b}_c$.

**The inpainting network.**  $\mathcal{I}$ is a 6 layer U-Net model with 8, 16, 32, 64, 128, 256 feature channels in each stage. The input to this network is an image where the selected bounding box region is hidden and the output is the entire image with the initially hidden patch being reconstructed. It is trained independently from the rest of the pipeline by feeding images with randomly occluded regions of varying sizes. In our full pipeline, the weights of the inpainting network are frozen and to remove the image evidence corresponding to the foreground person, the hidden patch in the input image to the inpainting network is selected to be bounding box region expanded by 20% in both dimensions.

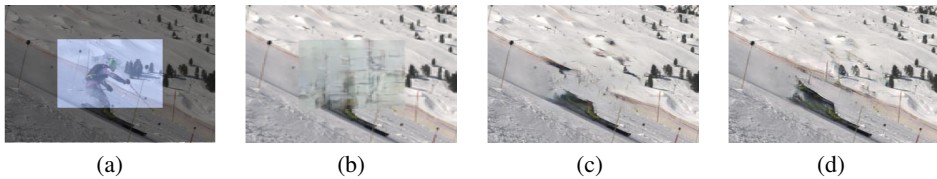

|     |     |     |     |
| --- | --- | --- | --- |
| (a) | (b) | (c) | (d) |

Figure 7: **Off-the-shelf inpainting results,** on skiing. (a) Input image with the hidden middle part, followed by inpainting with (b) Pathak et al. (2016), (c) Yu et al. (2018) trained on ImageNet. (d) and Yu et al. (2018) trained on Places2.

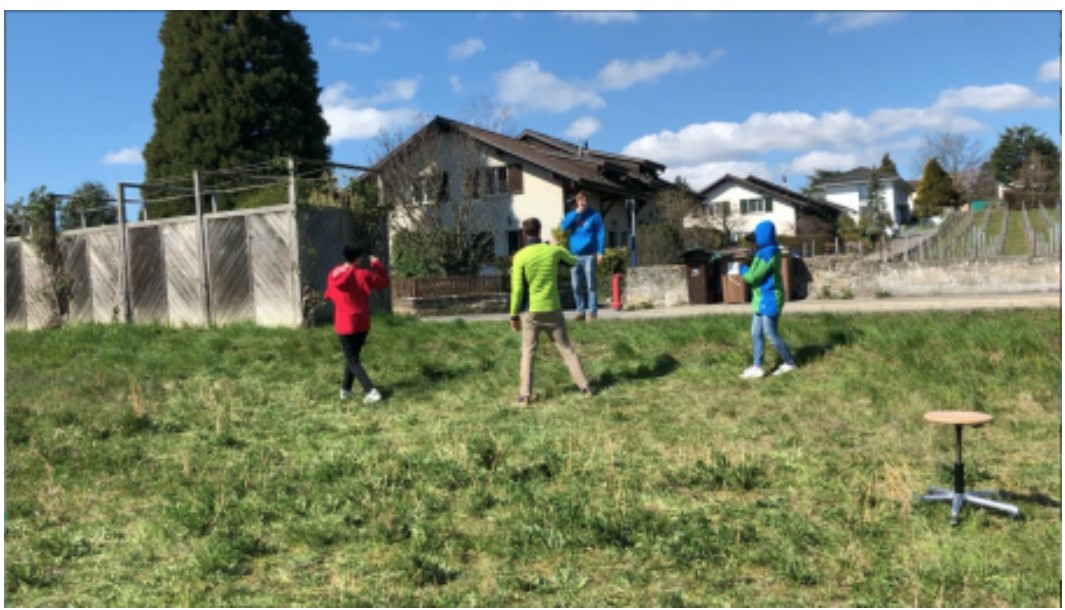

Figure 8: **The capture setup of our in-house handheld posing dataset. The subject is recorded by three persons with handheld GoPro action cameras.**

In principle, any off-the-shelf inpainting network trained on large and generic background datasets could be used. For instance, the methods of Pathak et al. (2016); Yu et al. (2018) can produce very plausible results. However, in domain-specific images, they tend to hallucinate objects as shown in Fig. 7 and are therefore ill-suited to our goal. Instead, we train $\mathcal{I}$ from scratch, by reconstructing randomly removed rectangular image regions, as shown in Fig. 2. Note that it is fine for this network not to generalize well to new scenes as it is not needed at test time.

**Overall training**   All training stages are performed on a single NVIDIA TITAN X (Pascal) GPU with Adam and a learning rate of 1e-3. First, the inpainting network is optimized for 200k iterations and subsequently the complete network for additional 100k iterations. The decoding pars of the synthesis network $\mathcal{S}$ uses a reduced learning rate of 1e-4, to prevent occasional diverging behavior. We use a batch size of 16 and an input image resolution of 640px×360px for the Ski-PTZ-camera dataset, 640px×360px for the in-house handheld posing dataset, and 500px×500px for H36M.

**Importance sampling**   For the importance sampling function $q$ we use $\epsilon = 0.001$, which makes the method numerically stable while the probability of choosing a random bounding box stays low, i.e., $6.4\%$ for 64 cells.

## A.2   Ski-PTZ-Dataset

In Table 3 we compare different mask priors and analyze the effect of using pre-trained weights. Although $\ell_1$ yields better segmentation masks than $\ell_2$, it might suppress the mask values too strictly which causes convergence problems. This is mitigated by our $\ell_v$ prior, which achieves the highest scores in all measures with consistently reliable results. Our unsupervised training method suffices for obtaining correct results from scratch. Nevertheless, given our relatively small dataset, using a pre-trained network significantly improves the stability of the training as well as the results. To this end we use an ImageNet pre-trained model since it is both practical

| | Ski-PTZ-Dataset | | | |
|---|---|---|---|---|
| Setting | Recall | Precision | F Measure | J Measure |
| No prior | 0.35 | 0.62 | 0.44 | 0.48 |
| $\ell_2$ prior | 0.43 | 0.64 | 0.51 | 0.47 |
| $\ell_1$ prior | 0.42 | 0.68 | 0.51 | 0.48 |
| $\ell_v$ prior | **0.52** | **0.77** | **0.62** | **0.56** |
| No Imagenet pre-training, $\ell_v$ prior | 0.36 | 0.55 | 0.43 | 0.46 |
| Unsupervised pre-training Wu et al. (2018), $\ell_v$ prior | 0.50 | 0.71 | 0.58 | 0.52 |

Table 3: **Analysis** of the mask prior effect and Imagenet pre-training on the **Ski-PTZ-Dataset** dataset. We demonstrate the influence of using mask priors to supress the noise surrounding the foreground object and have clear-cut masks. At the bottom part of the table we show the results of using random weights and features from Wu et al. (2018) instead of using weights from Imagenet pre-training.

| | Ski-PTZ-Dataset | | | |
|---|---|---|---|---|
| Setting | Recall | Precision | F Measure | J Measure |
| $\mathbf{b}_{[0.1,0.5]}, \ell_{v,\lambda=0.001}$ | **0.62** | 0.35 | 0.44 | 0.43 |
| $\mathbf{b}_{[0.1,0.5]}, \ell_{v,\lambda=0.0025}$ | 0.38 | 0.65 | 0.47 | 0.41 |
| $\mathbf{b}_{[0.1,0.5]}, \ell_{v,\lambda=0.005}$ | 0.37 | 0.65 | 0.47 | 0.43 |
| $\mathbf{b}_{[0.1,0.5]}, \ell_{v,\lambda=0.01}$ | 0.38 | 0.62 | 0.47 | 0.43 |
| $\mathbf{b}_{[0.15,0.5]}, \ell_{v,\lambda=0.001}$ | 0.35 | 0.59 | 0.43 | 0.43 |
| $\mathbf{b}_{[0.15,0.5]}, \ell_{v,\lambda=0.0025}$ | 0.50 | 0.69 | **0.57** | 0.51 |
| $\mathbf{b}_{[0.15,0.5]}, \ell_{v,\lambda=0.005}$ | 0.35 | 0.58 | 0.43 | 0.42 |
| $\mathbf{b}_{[0.15,0.5]}, \ell_{v,\lambda=0.01}$ | 0.39 | 0.65 | 0.48 | 0.46 |
| $\mathbf{b}_{[0.20,0.5]}, \ell_{v,\lambda=0.001}$ | 0.48 | **0.71** | **0.57** | **0.52** |
| $\mathbf{b}_{[0.20,0.5]}, \ell_{v,\lambda=0.0025}$ | 0.44 | 0.64 | 0.52 | 0.45 |
| $\mathbf{b}_{[0.20,0.5]}, \ell_{v,\lambda=0.005}$ | 0.37 | 0.63 | 0.46 | 0.44 |
| $\mathbf{b}_{[0.20,0.5]}, \ell_{v,\lambda=0.01}$ | 0.37 | 0.59 | 0.45 | 0.42 |
| $\mathbf{b}_{[0.25,0.5]}, \ell_{v,\lambda=0.001}$ | 0.39 | 0.58 | 0.46 | 0.43 |
| $\mathbf{b}_{[0.25,0.5]}, \ell_{v,\lambda=0.0025}$ | 0.40 | 0.62 | 0.48 | 0.44 |
| $\mathbf{b}_{[0.25,0.5]}, \ell_{v,\lambda=0.005}$ | 0.43 | 0.63 | 0.51 | 0.47 |
| $\mathbf{b}_{[0.25,0.5]}, \ell_{v,\lambda=0.01}$ | 0.41 | 0.61 | 0.49 | 0.45 |
| $\mathbf{b}_{[0.30,0.5]}, \ell_{v,\lambda=0.001}$ | 0.38 | 0.55 | 0.45 | 0.41 |
| $\mathbf{b}_{[0.30,0.5]}, \ell_{v,\lambda=0.0025}$ | 0.41 | 0.54 | 046 | 0.40 |
| $\mathbf{b}_{[0.30,0.5]}, \ell_{v,\lambda=0.005}$ | 0.39 | 0.54 | 0.45 | 0.42 |
| $\mathbf{b}_{[0.30,0.5]}, \ell_{v,\lambda=0.01}$ | 0.40 | 0.59 | 0.48 | 0.44 |
| $\mathbf{b}_{[0.35,0.5]}, \ell_{v,\lambda=0.001}$ | 0.50 | 0.64 | 0.56 | 0.51 |
| $\mathbf{b}_{[0.35,0.5]}, \ell_{v,\lambda=0.0025}$ | 0.44 | 0.62 | 051 | 0.46 |
| $\mathbf{b}_{[0.35,0.5]}, \ell_{v,\lambda=0.005}$ | 0.35 | 0.48 | 0.40 | 0.38 |
| $\mathbf{b}_{[0.35,0.5]}, \ell_{v,\lambda=0.01}$ | 0.40 | 0.51 | 0.45 | 0.40 |
| $\mathbf{b}_{[0.20,0.55]}, \ell_{v,\lambda=0.001}$ | 0.39 | 0.62 | 0.48 | 0.45 |
| $\mathbf{b}_{[0.20,0.60]}, \ell_{v,\lambda=0.001}$ | 0.48 | 0.68 | 0.56 | 0.49 |
| $\mathbf{b}_{[0.20,0.65]}, \ell_{v,\lambda=0.001}$ | 0.38 | 0.61 | 0.46 | 0.46 |
| $\mathbf{b}_{[0.20,0.70]}, \ell_{v,\lambda=0.001}$ | **0.54** | 0.73 | **0.62** | **0.56** |
| $\mathbf{b}_{[0.20,0.75]}, \ell_{v,\lambda=0.001}$ | **0.54** | 0.74 | **0.62** | 0.55 |
| $\mathbf{b}_{[0.20,0.80]}, \ell_{v,\lambda=0.001}$ | 0.52 | **0.77** | **0.62** | **0.56** |

Table 4: **Hyper-parameter study** on the **Ski-PTZ-Dataset** dataset. In this table we analyze the effectiveness of our hyper-parameter choice for the minimum and maximum bounding box sizes (given in square brackets) as well as the threshold $\lambda$ for the $\ell_v$ loss.

and the weights are readily available for diverse architectures, although from a different domain and task. To quantify the reliability of our method with the $\ell_v$ regularizer, we repeated the **Ski-PTZ-Dataset** experiment three times with the best-performing configuration and computed the mean and std; the Precision, Recall, F- and J-Measure are consistent, respectively, $0.54 \pm 0.02, 0.60 \pm 0.01, 0.74 \pm 0.02, 0.51 \pm 0.01$.

In Table 4 we compare the performance of our method for different values of hyper-parameters, where the subscript of $\mathbf{b}$ corresponds to the minimum and maximum size of the bounding box and $\lambda$ is defined as in Eq. (7).

$$L_v = \left| \left( \frac{1}{WH} \sum_x^W \sum_y^H \mathcal{T}^{-1}(\mathbf{S})_{xy} \right) - \lambda \right| + \lambda \, , \tag{7}$$

## A.3 **Handheld190k** DATASET

The capture setup of our **Handheld190k** dataset is depicted in Fig. 8. The dataset is composed of five actors performing the same actions as those available in the H36M dataset Ionescu et al. (2014), namely *directions, discussion, eating, greeting, phone talk, posing, buying, sitting, sitting down, smoking, taking photo, waiting, walking, walking dog* and *walking in pair*. We excluded *lying on the floor* actions, to not making our actors lie in the dirt. The data from three actors compose our training set and the other two form the test set. The data was obtained using 3 GoPro6 cameras recording FullHD videos at 30 FPS in linear lens mode. As seen in the Fig. 8 it is an outside recording. For the entire duration, the cameras were subject to lateral movement and varying hand-held rotation. The motion stablization of the GoPros was deactivated at the time of the recording.

## A.4 QUALITATIVE RESULTS

We provide additional examples of our detection and segmentation results on the **Handheld190k** test set in Fig. 9. Fig. 9 also shows results from the ski test set. Additional segmentations on the established **H36M** dataset are given in Fig. 10. Notably, our method is robust to the undirected camera motion and to dynamic background motion, and works equally well for the very different domains of skiing and every-day activities.

## A.5 IMPORTANCE SAMPLING THEORY

In the following, we give an explanation for the change of distribution when computing an expectation. Let $p, q$ be two discrete probability distributions and $f(c)$ an arbitrary function of $c$. Reasoning about the limit towards infinitely many samples $c_j$ drawn from $q(\cdot)$, we derive

$$\mathbf{E}_{q(c)} \left[ \frac{p(c)}{q(c)} f(c) \right] = \lim_{J \to \infty} \frac{1}{J} \sum_{j=1}^{J} \left( \frac{p(c_j)}{q(c_j)} f(c_j) \right)$$

$$= \lim_{J \to \infty} \frac{1}{J} \sum_{c=1}^{C} \sum_{k=1}^{\sum_{j=1}^{J} 1_{c_j=c}} \left( \frac{p(c)}{q(c)} f(c) \right)$$

$$= \frac{1}{J} \sum_{c=1}^{C} J q(c) \left( \frac{p(c)}{q(c)} f(c) \right)$$

$$= \sum_{c=1}^{C} \left( p(c) f(c) \right)$$

$$= \mathbf{E}_{p(c)} \left[ f(c) \right] \tag{8}$$

where $J$ is the number of samples drawn and $1_{c_j=c}$ is one if $c_j$ equals $c$. In the second line, we exploit that we have a finite number of classes $C$. Each sample must fall into one of them and the probability of coming from cell $c$ is $q(c)$.

This relation provides us with a tool to change the sample distribution for expectations. Next, we analyze te variance of such estimator depending on the chosen sampling distribution.

## A.6 IMPORTANCE SAMPLING VARIANCE

The variance of an estimator gives us a measure for the expected accuracy after a limited number of samples. We use it to compare the three candidates discussed in the main document. The variance of estimating the objective $O$ with a monte carlo sum over $c_1, \ldots, c_J$ samples drawn independently from $p$ is

$$\text{Var}\left[\mathcal{O}(\mathbf{I})\right] \approx \text{Var}\left[ \frac{1}{J} \sum_{j=1}^{J} L\left(\mathcal{F}(\mathbf{I}, c_j), \mathbf{I}\right) \right]$$

$$= \frac{1}{J^2} \sum_{j=1}^{J} \text{Var}\left[ L\left(\mathcal{F}(\mathbf{I}, c_j), \mathbf{I}\right) \right]$$

$$= \frac{1}{J} \text{Var}\left[ L\left(\mathcal{F}(\mathbf{I}, c_j), \mathbf{I}\right) \right] \, . \tag{9}$$

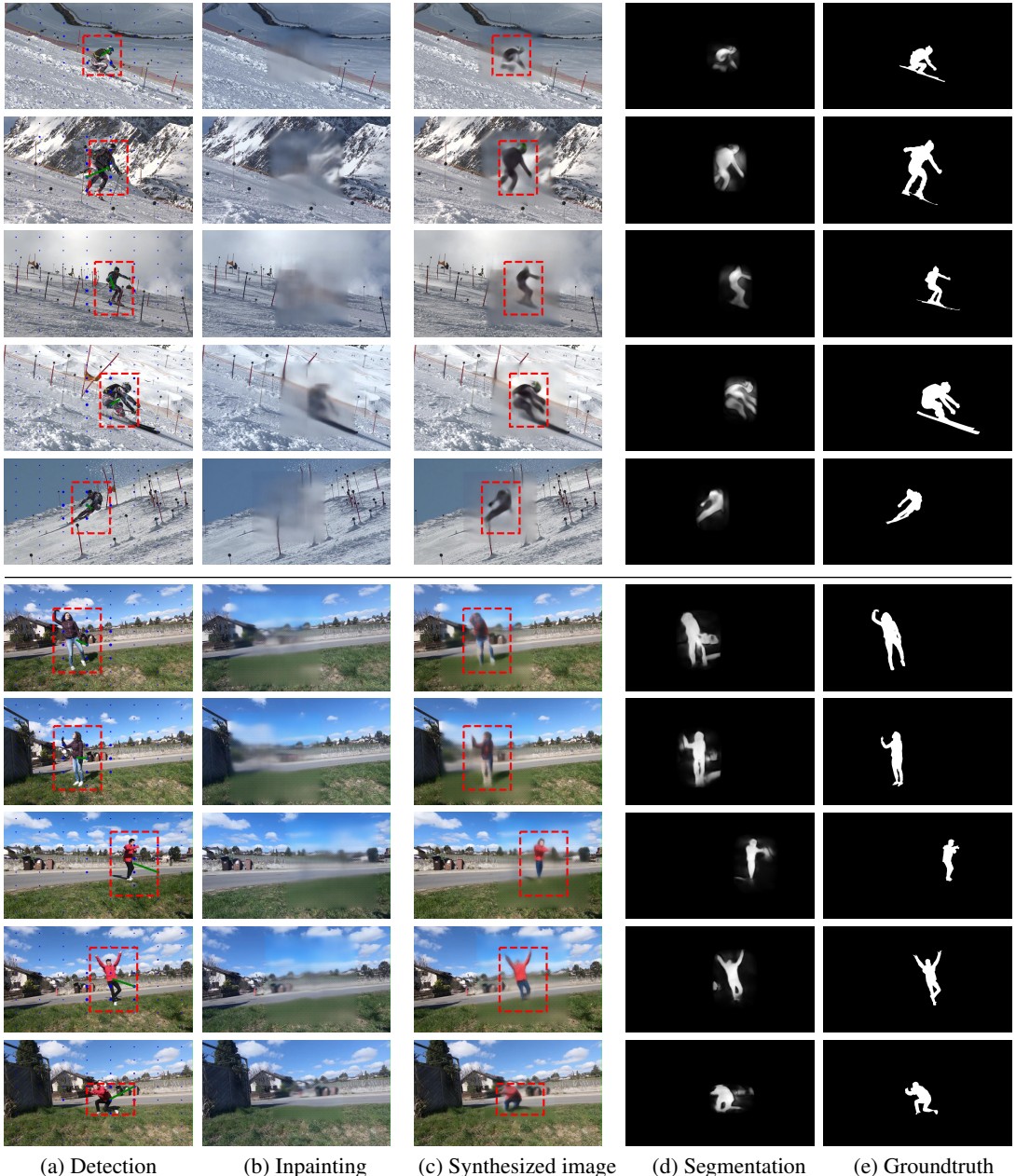

(a) Detection  (b) Inpainting  (c) Synthesized image  (d) Segmentation  (e) Groundtruth

Figure 9: **Detection and segmentation results on the test subjects of Ski-PTZ-camera and Hand-held190k dataset.** (a) Detection result. The blue dots coincide with the grid cell centers and their size indicates the confidence of the bounding box proposals. The selected bounding box is illustrated with a red dashed line and the center of the grid cell yielding this proposal is connected to the center of the red box through the green line. (b) The inpainting result where the region inside the detected bounding box is reconstructed by our inpainting network (only needed for training). (c) The synthesized image with the predicted foreground and background regions combined. (d) Segmentation mask result. (e) Groundtruth segmentation mask.

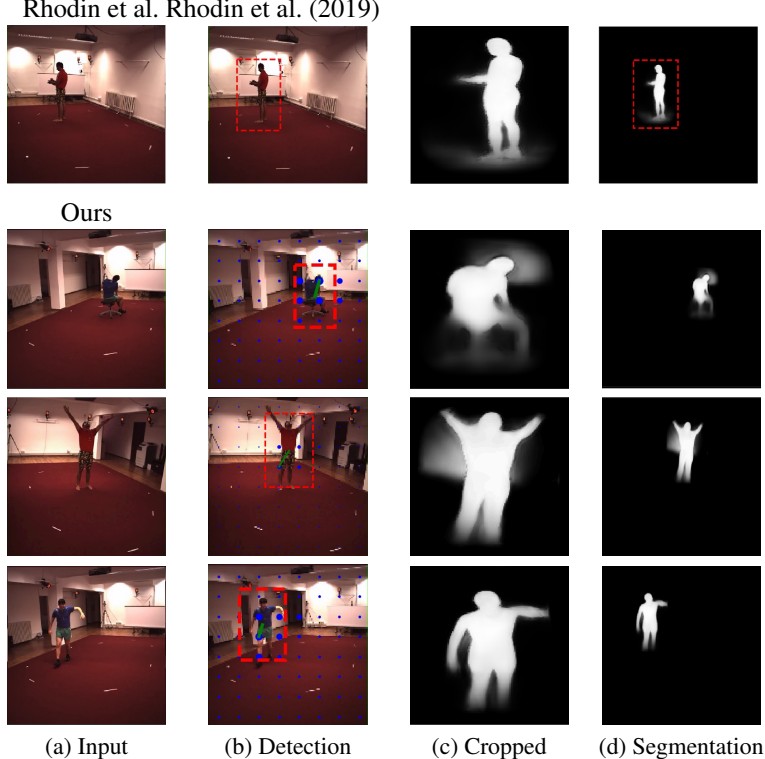

(a) Input       (b) Detection       (c) Cropped       (d) Segmentation

Figure 10: **Detection and segmentation results on H36M.** Results match in quality with those from Rhodin et al. (2019), with a slight bleeding due to not having a perfect background prediction oracle.

Here $\mathrm{Var}\left[L\left(\mathcal{F}(\mathbf{I}, c_j), \mathbf{I}\right)\right]$ is the variance over the random variable $c_j$, and we utilized the identity $\mathrm{Var}[ax] = a^2 \mathrm{Var}[x]$ and the independence of samples. The variance reduces linearly with the number of samples and is proportional to that of $L(\cdot)$.

In contrast, using uniform sampling (Eq. 4, main document, with $q = U_c$), yields a quadratic variance growth with the number of cells,

$$
\begin{aligned}
\mathrm{Var}\left[\mathcal{O}(\mathbf{I})\right] &\approx \frac{1}{J} \mathrm{Var}\left[\frac{p(c_j)}{\mathcal{U}_C(c_j)} L\left(\mathcal{F}(\mathbf{I}, c_j), \mathbf{I}\right)\right] \\
&= \frac{1}{J} \mathrm{Var}\left[Cp(c_j) L\left(\mathcal{F}(\mathbf{I}, c_j), \mathbf{I}\right)\right] \\
&= \frac{C^2}{J} \mathrm{Var}\left[p(c_j) L\left(\mathcal{F}(\mathbf{I}, c_j), \mathbf{I}\right)\right] .
\end{aligned}
\tag{10}
$$

From which we conclude that uniform sampling leads to a higher variance, since the remaining $\mathrm{Var}[p(\cdot)L(\cdot)]$ term is not expected to improve on $\mathrm{Var}[L(\cdot)]$.

With importance sampling according to $q$ (Eq. 4, main document) the variance is,

$$
\mathrm{Var}\left[\mathcal{O}(\mathbf{I})\right] \approx \frac{1}{J} \mathrm{Var}\left[\frac{p(c)}{q(c)} L\left(\mathcal{F}(\mathbf{I}, c), \mathbf{I}\right)\right]
\tag{11}
$$

If $q \approx p$ it is equivalent to the one of Eq. 9. In general, $q$ should be constructed to minimize Eq. 11. In our case, $\mathcal{F}$ depends on each individual image and cell $c$, it is thereby difficult to impose assumptions to reduce the variance of $L$ without evaluating it for each $c$. Therefore, setting $q \approx p$ is a good choice from the perspective of variance reduction.

## A.7 ADDITIONAL COMPARISONS TO RELATED WORK

Motion-based approaches are prone to fail when there is no or too complex motion information to separate foreground and background and in textureless areas. These failure cases occur in all our scenarios, as shown in the optical flow images in Fig. 11 generated from our datasets. To this extend, we excluded motion cues in this study. We are planning on integrating their complementary merits in the future.

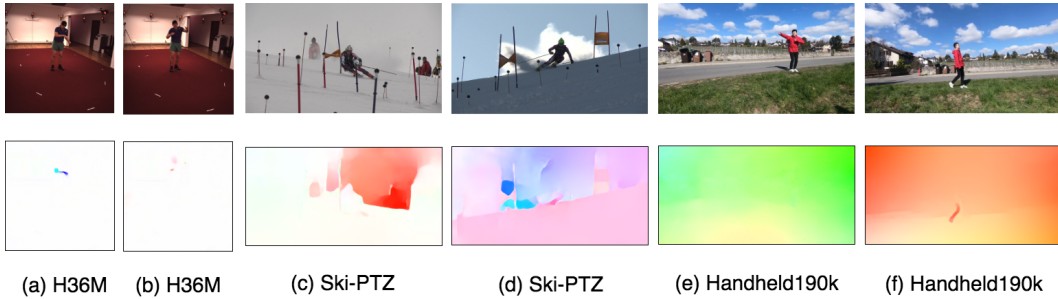

(a) H36M    (b) H36M    (c) Ski-PTZ    (d) Ski-PTZ    (e) Handheld190k    (f) Handheld190k

Figure 11: **Optical flow failure cases. Top:** Input images with no or too complex motion information. **Bottom:** Corresponding optical flow images generated using Ilg et al. (2017). The white portions in the flow show the regions that are stationary relative to the camera. Multiple colors in the flow indicate different motion directions which can not be easily segmented into foreground and background.

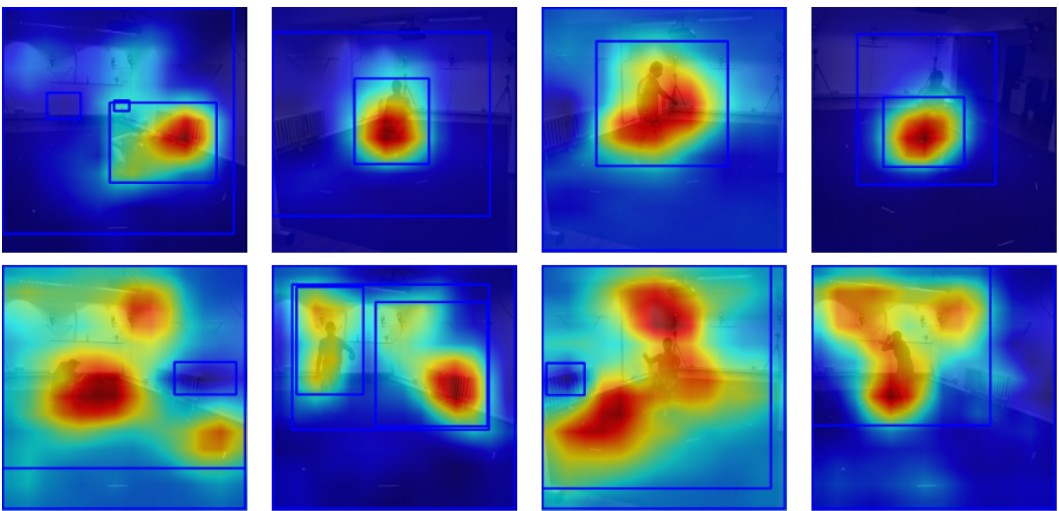

Figure 12: **Examples of heatmaps and detection results from Zhou et al. (2016) on H36M.** The heatmap generated by the ImageNet pre-trained model has multiple high activations corresponding to a given ImageNet category. Since the candidate bounding boxes are selected such as to overlay the connected components of the heatmap, they can cover a large area. Therefore using these candidates to efficiently localize the salient object can often fail.

We also investigate the possibility of using a pre-trained localization network from Zhou et al. (2016) and obtain the heatmaps and the corresponding bounding box predictions for H36M S9 and S11 test images using their GoogLeNet-CAM pre-trained model on ImageNet. As the output heatmaps in Fig. 12 show, the pre-training on ImageNet alone is not reliable to have accurate detections. This is mainly due to the multiple activated regions that yield very big bounding boxes covering not just the human subject but also the other objects in the scene matching with the ImageNet classes. Among all the bounding box predictions that Zhou et al. (2016) generates, we select the one yielding the highest IOU score against the groundtruth bounding box corresponding to that frame. We report that Zhou et al. (2016) has an $mAP_{0.5}$ score of 0.1 which is significantly lower than our $mAP_{0.5}$ score of 0.58. This gap indicates that doing the detection using only their pre-trained model is not reliable and selection of the best bounding box among the candidates is unfeasible in a self-supervised setting. Our model can accurately detect the human subject without needing an ImageNet pre-training in the detection component. Regarding the pre-training on the encoder part of $\mathcal{S}$ we refer the reader to Table 3. Comparing the ImageNet pre-trained model (in the fourth row) with the unsupervised pre-trained model (in the last row), the unsupervised setup shows only a small performance drop; pre-training helps but is not essential.

Another part of our approach where we can provide additional comparison is the sampling method. To explore an alternative strategy to Monte-Carlo based sampling, we tried replacing the importance sampling part in our method with the categorical reparameterization used in Crawford & Pineau (2019). Since both strategies approximate the same objective, they should lead to very similar outcomes with a possible difference in the

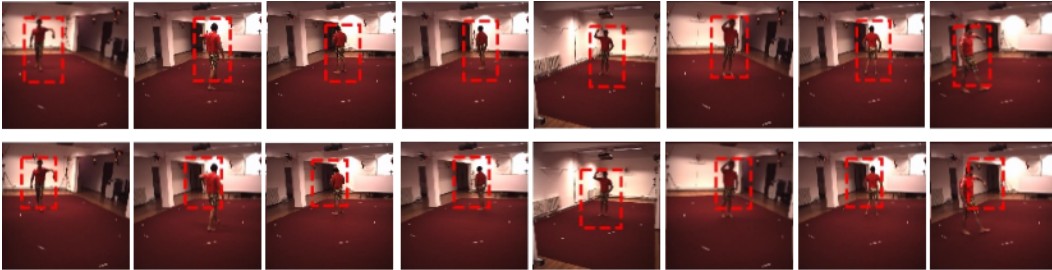

Figure 13: **Qualitative results comparing the bounding box detections obtained through our importance sampling strategy against the Gumbel-Softmax categorical reparameterization with temperature hyperparameter set to** 0.1**. Top:** Ours. **Bottom:** Categorical reparameterization. The results are taken from the same iteration for both models.

convergence speed. To this end we used Gumbel-Softmax distribution and tried this estimator with several different temperature values. Our experiments show that Gumbel-Softmax based categorical reparameterization didn't lead to faster convergence as shown in Fig. 13. It can be seen that for the same iteration from which the results are taken the importance sampling based model has already converged in terms of detection whereas the categorical reparameterization hasn't converged yet. This might be partly due to the high value of Gumbel noise added to the log probabilities. However, to have a more solid claim, a principled grid search for hyperparameters such as the temperature is necessary. We will provide the exact comparison in our final version after conducting an extensive hyperparameter search. Overall our importance sampling approach is simpler compared to their approach and has the advantage of being an unbiased estimator. In addition to that it does not need custom layers that behave differently in the forward and backwards passes during optimization, which is the case for the Gumbel-Softmax categorical reparameterization.

