# OpenReview forum: "Self-supervised Training of Proposal-based Segmentation via Background Prediction"
_ICLR.cc/2020/Conference — Reject_

### Official Review · AnonReviewer3 · 2019-10-20
**Official Blind Review #3**

**Rating:** 6

**Review:**

This submission proposes a self-supervised segmentation method, that learns from single-object videos by finding the region where it can segment an object, remove the entire bounding box around it, inpaint it, then finally put the object back. The loss is a balance between a reconstruction error and a negative inpainting error (high error means object is probably present, due to the weak correlation with the background).

My decision is Weak Accept. I like the method very much and think it’s a clever and well-executed algorithm. The reason for being Weak is because the experimental evidence could be stronger, especially comparing with Croitoru et al. and Rhodin et al. The paper leaves some open problems, inviting future work to be built on top of it (i.e. leveraging time more and handling multiple objects better). I think it should be accepted to promote such future work.

Method:

The method is interesting and clever. Similar efforts have been made, such as Bielski & Favaro and Crawford & Pineau. However, key contributions relax some of the contrived requirements of these past methods (e.g. simple foreground translation over background; requirement of plain background). Thanks to the inpainter, the importance sampler, and avoidance of collapsing into trivial solutions, this paper is able to put together a method that works on moving cameras and fairly complex scene semantics (still limited to one object though). Actually getting this to converge with only two loss terms balanced against each other is impressive. Having certain heads of the network trained only on one of the two terms seems to be a key contribution.

Experimental results:

The comparison with Rhodin et al. on H36M is characterized as “slightly” lower, but I would call 71% vs. 58% a significant difference. Of course, I understand that Rhodin et al. relies on the static background, so this is not a fair comparison.

Ski-PTZ-Dataset should then offer a better comparison, but here the method struggles to compete with Croitoru et al. 2019, which has a 11-point higher F-measure. On Handheld190k, the dataset proposed in this paper, is where the method finally shines, but still only offers a 1-point F-measure improvement over Croitoru et al. That being said, considering how different the methods are, and how Croitoru et al. requires two-stage training, there are many benefits to this method. Croitoru et al. also relies on video to extract the object features, and this requirement is not as explicit in this work. Actually, that brings me to one question I had. The paper states that “as long as videos or picture collections of a single object in front of the same scene are available.” I didn’t quite understand why this must be trained on multiple images of the same scene. If the inpainter is general to any background scenery, couldn’t it work on single-images as well? The conclusion even says you do not use temporal cues.

Other:

I don’t think it’s that meaningful to include precision/recall in the tables. It is also not that meaningful to point out that your method’s precision is higher than that of Croitoru et al., when the F-measure is shy 11 points. The reason is because many points on the precision/recall could be constructed simply by applying for instance a gamma curve on the segmentation predictions. The high precision is clearly at the cost of a low recall, and another point on this tradeoff curve could be presented. This is why F-measure and average precision are much better.

**Experience Assessment:**

I have published one or two papers in this area.

**Review Assessment: Checking Correctness Of Derivations And Theory:**

I carefully checked the derivations and theory.

**Review Assessment: Checking Correctness Of Experiments:**

I carefully checked the experiments.

**Review Assessment: Thoroughness In Paper Reading:**

I read the paper at least twice and used my best judgement in assessing the paper.

---

> ### Author Response · Authors · 2019-11-13
> **Initial Response to Review #3**
>
> Thank you for the constructive feedback. We are glad that you find the idea original. We address your concerns below.
>
> The changes in the main text are highlighted in red.
>
> Reviewer 3: Precision/recall scores
>
> We agree with your point on the compromise between precision and recall. Therefore, we will rewrite our comparison to focus on J and F measures; precision and recall will only be used to explain where additional improvements are needed.
>
> Reviewer 3: Training of multiple images of the same scene
>
> This comment is related to Reviewer 1’s: “… assumption that the background follows relatively consistent textures”. Therefore, we restate our response here.
>
> Indeed, using an off-the-shelf inpainting network performed poorly because our images differ significantly from those in the datasets it had been trained on.
> Therefore, we trained our own inpainting model (see Page 11, Section A1 Implementation Details, The inpainting network). It can deal with diverse backgrounds as well as non-consistent ones given enough data. Although in the Ski-PTZ dataset the scene looks homogeneous, in HandHeld190k the scene is cluttered with different textured objects such the houses, fences and trees. It requires multiple images of the same scene, or videos. Note, that we can leverage the input videos directly; we don’t need additional background images without persons. In the revised version we added a more detailed discussion about the inpainting network.

---

### Official Review · AnonReviewer2 · 2019-10-22
**Official Blind Review #2**

**Rating:** 3

**Review:**

The paper introduces a method to self-supervised train a model for object detection/segmentation. The idea is that the background is easy to reconstruct while the foreground/object is hard. Experiments demonstrate the effectiveness of the proposed methods.

Here are some high-level concerns.

1. As mentioned in the "Implementation details", 'naive end-to-end training is difficult... we use ImageNet-trained weights for initialization'. This is worrisome to justify the effectiveness. It may be possible the imagenet-trained model has already captured salient objects. To justify the claims and effectiveness of the method, it should include a comparison with [R1], which demonstrates the possibility of doing detection with a pretrained model. Other work along this line should be also good reference.

2. As a moving camera is available, it is also possible to segment background with frames through a 6DoF prediction on the camera, rotation and translation, e.g., [R2]. The supervision signal is from frame reconstruction through learning to predict both camera pose and pixel-level depth. This is also self-supervised learning. At least such a self-supervised trained model can act as an initialization.

Considering the above points, the paper does not appear compelling, due to lack of either careful claims or justification.


[R1] Learning deep features for discriminative localization
[R2] Unsupervised learning of depth and ego-motion from video

**Experience Assessment:**

I have published one or two papers in this area.

**Review Assessment: Checking Correctness Of Derivations And Theory:**

I did not assess the derivations or theory.

**Review Assessment: Checking Correctness Of Experiments:**

I assessed the sensibility of the experiments.

**Review Assessment: Thoroughness In Paper Reading:**

I read the paper at least twice and used my best judgement in assessing the paper.

---

> ### Author Response · Authors · 2019-11-13
> **Initial Response to Review #2**
>
> Thank you for your constructive comments. We address your concerns in detail below.
>
> The changes in the main text are highlighted in red.
>
> Reviewer 2: Comparison to [R1] “Learning deep features for discriminative localization” and ImageNet pre-training
>
> The only component where we use a pre-trained model is the ResNet50 module in the encoder component of S (see Page 11, Section A1 Implementation Details, The synthesis network). This component does not perform localization, and only encodes the selected patch of the image. In the last two rows of Table 3 in the Appendix we report the performance of our model using a randomly initialized ResNet and also an unsupervised pre-trained model (from Wu et al. 2018). Comparing the ImageNet pre-trained model (in the fourth row) with the unsupervised pre-trained model (in the last row), the unsupervised setup shows only a small performance drop; pre-training helps but is not essential.
>
> We also compare the detection accuracy of [R1] in the revised  version, note, however, that [R1] does not provide accurate segmentation masks but rather heatmaps.
>
> To compare our method to [R1] we first obtain the heatmaps and the corresponding bounding box predictions for H36M S9 and S11 test images using the official code and their GoogLeNet-CAM pre-trained model on ImageNet. As the output heatmaps of [R1] illustrate (see page 17, section A7 Additional Comparisons to Related Work, Fig. 12), there can be several highly activated regions in the heatmaps leading to very big bounding boxes. In some cases, the model highlights other objects in the scene since ImageNet contains a wide range of object classes. Among all the bounding box predictions that [R1] generates, we select the one yielding the highest IOU score against the ground-truth bounding box corresponding to that frame. We report that [R1] has an mAP_{0.5} score of 0.1 which is significantly lower than our mAP_{0.5} score of 0.58. This gap indicates that performing detection using only their pre-trained model is not reliable and selection of the best bounding box among the candidates is infeasible in a self-supervised setting. Our model can accurately detect the human subject without needing ImageNet pre-training in the detection component.
>
> Reviewer 2: Comparison to [R2] “Unsupervised learning of depth and ego-motion from video” and using motion as the supervision.
>
> This comment is related to Reviewer 1’s: “optical flow and boundary detection, which I thought are OK cues to be used”. Therefore, we restate our response here.
>
> While one can use optical flow and other motion cues from unsupervised techniques to boost performance, the independence of such cues makes our proposed approach applicable to single images and, most importantly, avoids failure cases of optical flow. Motion-based approaches are prone to failure when there is no or too complex motion information to separate the foreground and background and in textureless areas. These failure cases occur in all our scenarios, as shown in the attached optical flow images (see page 17, section A7 Additional Comparisons to Related Work, Fig. 11) generated from our datasets  using FlowNet2.0. Depth and ego-motion prediction suffers from similar problems. There are always multiple ways of addressing the same problem and only in retrospect the preferred strategy becomes clear. In this study, we investigated how far (quite far!) one can get without motion cues. Nevertheless, as we state in the outlook section, we will combine the complementary merits of motion-based strategies in the future; using [R2] will be a viable addition.

---

### Official Review · AnonReviewer1 · 2019-10-28
**Official Blind Review #1**

**Rating:** 3

**Review:**

This paper provides a new self-supervised proposal-based approach for object detection and segmentation. The author introduces a Monte Carlo-based optimization to solve the inefficiency problem in the discrete proposal-based forward process defined in (Crawford and Pineau 2019). Also, the paper redefines the decoder part for self-supervised from minimizing reconstruction loss with background segmentation to maximize reconstruction error with learning a foreground segmentation.  The method is then verified with a suite of experiments for people-detection on video datasets.

The main benefit over many previous unsupervised object detection/segmentation approaches is that they did not make use of optical flow or other readily available cues during training. However, given that the framework directly came from (Crawford & Pineau 2019), and the only change is from variational inference to an importance-sampling (MC) approach. This would be fine if it is verified in experiments, however, the experiments did not show any comparison w.r.t. (Crawford & Pineau 2019) hence we have no way of understanding what is the relative performance w.r.t. that baseline approach.

Besides, in all the experiments a single object is in the view. How does the method perform in images where multiple objects are in the view?

A little bit of a philosophical question is whether this a problem worth pursuing as well. For self-supervised motion estimation (e.g. optical flow), it is clear why we want to do that. However, the current type of algorithm is dependent on the assumption that the background follows relatively consistent textures, this may not necessarily be true in practice, and hence the application could be quite limited. Many previous unsupervised video object segmentation methods make use of optical flow and boundary detection, which I thought are OK cues to be used, especially when both can be learned in a self-supervised manner. This is not entirely related to the assessment, but I would still like to hear what the authors think.

Minor:
In the paragraph after 'Training strategy'(Section 3.2, Page 5), is it 'the foreground objective O of Eq. (2)' or 'Eq.(4)'?



**Experience Assessment:**

I have published one or two papers in this area.

**Review Assessment: Checking Correctness Of Derivations And Theory:**

I assessed the sensibility of the derivations and theory.

**Review Assessment: Checking Correctness Of Experiments:**

I assessed the sensibility of the experiments.

**Review Assessment: Thoroughness In Paper Reading:**

I read the paper at least twice and used my best judgement in assessing the paper.

---

> ### Author Response · Authors · 2019-11-13
> **Initial Response to Review #1**
>
> Thank you for your insightful comments. We address your concerns in detail below.
>
> The changes in the main text are highlighted in red.
>
> Reviewer 1: “… the framework directly came from (Crawford & Pineau 2019), and the only change is from variational inference to an importance-sampling (MC) approach”
>
> We would like to clarify that the novelty of our method with respect to (Crawford & Pineau 2019) does not only come from the Monte-Carlo based strategy for handling the discrete proposals through an unbiased estimator. In addition to changing from a variational inference using Gumbel-Softmax to a Monte-Carlo sampling approach, the model in (Crawford and Pineau 2019) is designed for multi-object detection in images with monochrome background, while our model tackles single object detection and segmentation in non-static backgrounds, making our approach easily applicable to outdoor scenes captured with moving cameras. Considering this, a direct comparison of the two models is not possible. In particular, there is no proposed mechanism in (Crawford and Pineau 2019) to deal with non-static backgrounds, as acknowledged in their paper. By contrast, the inpainting model in our approach, through the proposal based bounding box selection framework, favors the locations that have high inpainting loss, thus guiding us to the salient object region in non-static backgrounds. We would like to point out that the key novelty here is casting the segmentation problem as an inpainting and search task which has not been attempted before. To this end, we come up with an efficient training strategy that allows a stable training using the adversarial foreground and background losses.
>
> A different comparison can be made by replacing the importance sampling part in our method with the categorical reparameterization used in (Crawford and Pineau 2019), which we will report before the end of the rebuttal period.
>
> Reviewer 1: “How does the method perform in images where multiple objects are in the view?”
>
> While we show in Fig. 4 of the paper that our model can tackle multi-object detection and segmentation by sampling more than once at test time, multi-object detection is not the focus of our current work. The algorithm can be extended for multi-object detection and segmentation, but since it was out of the scope of the current model, we did not pursue that particular line of research.
>
> Reviewer 1: “… assumption that the background follows relatively consistent textures”
>
> Indeed, using an off-the-shelf inpainting network performed poorly because our images differ significantly from those in the datasets it had been trained on.
> Therefore, we trained our own inpainting model (see Page 11, Section A1 Implementation Details, The inpainting network). It can deal with diverse backgrounds as well as non-consistent ones given enough data. Although in the Ski-PTZ dataset the scene looks homogeneous, in HandHeld190K the scene is cluttered with different textured objects such the houses, fences and trees. It requires multiple images of the same scene, or videos. Note, that we can leverage the input videos directly; we don’t need additional background images without persons.
>
> Reviewer 1: “optical flow and boundary detection, which I thought are OK cues to be used”
>
> While one can use optical flow and other motion cues from unsupervised techniques to boost performance, the independence of such cues makes our approach applicable to single images and, most importantly, avoids failure cases of optical flow. Motion-based approaches are prone to failure when there is no or too complex motion information to separate the foreground and background and in textureless areas. These failure cases occur in all our scenarios, as shown in the attached optical flow images (see page 17, section A7 Additional Comparisons to Related Work, Fig. 11) generated from our datasets using FlowNet2.0. Depth and ego-motion prediction suffers from similar problems. There are always multiple ways of addressing the same problem and only in retrospect the preferred strategy becomes clear. In this study, we investigated how far (quite far!) one can get without motion cues. Nevertheless, as we state in the outlook section, we will combine the complementary merits of motion-based strategies in the future; using [R2, mentioned by Reviewer 2] will be a viable addition.
>
> Reviewer 1: Typo in Section 3.2 page 5
>
> Thanks for pointing out the wrong equation number. It should be “the foreground objective O of Eq. (4)”.

---

> > ### Author Response · Authors · 2019-11-15
> > **Categorical reparameterization vs importance sampling**
> >
> > We tried replacing the importance sampling part in our method with the categorical reparameterization used in (Crawford and Pineau 2019). Since both strategies approximate the same objective, they should lead to very similar outcomes with a possible difference in the convergence speed. To this end we used Gumbel-Softmax distribution and tried this estimator with several different temperature values. Our experiments show that Gumbel-Softmax based categorical reparameterization didn’t lead to faster convergence (see page 18 in SectionA7, Fig. 13). This might be partly due to the high value of Gumbel noise added to the log probabilities. However, to have a more solid claim, a principled grid search for hyperparameters such as the temperature is necessary. We will provide the exact comparison in our final version after conducting an extensive hyperparameter search.
> >
> > Overall our importance sampling approach is simpler compared to their approach and has the advantage of being an unbiased estimator. In addition to that it does not need custom layers that behave differently in the forward and backwards passes during optimization, which is the case for the Gumbel-Softmax categorical reparameterization.

---

### Decision · Program_Chairs · 2019-12-19

**Decision:**

Reject

**Comment:**

This work proposes a self-supervised segmentation method: building upon Crawford and Pineau 2019, this work adds a Monte-Carlo based training strategy to explore object proposals.
Reviewers found the method interesting and clever, but shared concerns about the lack of a better comparison to Crawford and Pineau, as well as generally a lack of care in comparisons to others, which were not satisfactorily addressed by authors response.
For these reasons, we recommend rejection.